# Vaccines’ New Era-RNA Vaccine

**DOI:** 10.3390/v15081760

**Published:** 2023-08-18

**Authors:** Wenshuo Zhou, Linglei Jiang, Shimiao Liao, Feifei Wu, Guohuan Yang, Li Hou, Lan Liu, Xinping Pan, William Jia, Yuntao Zhang

**Affiliations:** 1CNBG-Virogin Biotech (Shanghai) Co., Ltd., Shanghai 201800, China; zhouwenshuo@cnbg-virogin.com (W.Z.); jianglinglei@cnbg-virogin.com.cn (L.J.); liaoshimiao@cnbg-virogin.com (S.L.); wufeifei@cnbg-virogin.com.cn (F.W.); ghyang@cnbg-virogin.com.cn (G.Y.); houli@cnbg-virogin.com (L.H.); liulan@cnbg-virogin.com (L.L.); panxp@cnbg-virogin.com (X.P.); wjia@virogin.com (W.J.); 2Shanghai-Virogin Biotech Co., Ltd., Shanghai 201800, China; 3Sinopharm Group China National Biotech Group (CNBG) Co., Ltd., Beijing 100124, China

**Keywords:** mRNA vaccine, circular RNA vaccine, saRNA vaccine, infectious diseases, drug delivery system

## Abstract

RNA vaccines, including conventional messenger RNA (mRNA) vaccines, circular RNA (circRNA) vaccines, and self-amplifying RNA (saRNA) vaccines, have ushered in a promising future and revolutionized vaccine development. The success of mRNA vaccines in combating the COVID-19 pandemic caused by the SARS-CoV-2 virus that emerged in 2019 has highlighted the potential of RNA vaccines. These vaccines possess several advantages, such as high efficacy, adaptability, simplicity in antigen design, and the ability to induce both humoral and cellular immunity. They also offer rapid and cost-effective manufacturing, flexibility to target emerging or mutant pathogens and a potential approach for clearing immunotolerant microbes by targeting bacterial or parasitic survival mechanisms. The self-adjuvant effect of mRNA-lipid nanoparticle (LNP) formulations or circular RNA further enhances the potential of RNA vaccines. However, some challenges need to be addressed. These include the technology’s immaturity, high research expenses, limited duration of antibody response, mRNA instability, low efficiency of circRNA cyclization, and the production of double-stranded RNA as a side product. These factors hinder the widespread adoption and utilization of RNA vaccines, particularly in developing countries. This review provides a comprehensive overview of mRNA, circRNA, and saRNA vaccines for infectious diseases while also discussing their development, current applications, and challenges.

## 1. Introduction

### 1.1. Conventional mRNA Vaccine

Over the past few decades, mRNA vaccine development has achieved significant milestones since the discovery of mRNA in 1961 [1]. Within eight years, the successful production of proteins from isolated mRNA in laboratory settings marked a critical advancement. Simultaneously, the lipid-drug delivery system, which started in 1965 with the creation of liposomes, was progressively employed in drug and vaccine delivery, respectively, in 1971 and 1974. Notably, a more refined drug delivery system utilizing four-component lipid nanoparticles was developed in 2001 and subsequently utilized for testing mRNA vaccines in mice, as well as clinical trials for rabies and influenza in 2013 and 2015, respectively [1]. In 2018, the first drug-employing lipid nanoparticles, patisiran, received approval. Furthermore, in response to the SARS-CoV-2 pandemic, mRNA-based COVID-19 vaccines were granted emergency authorization by the FDA in 2020, effectively safeguarding individuals against SARS-CoV2 infection. The advancement of mRNA vaccine technology has fostered the establishment of several mRNA vaccine companies. Merix Biosciences, initially known as Argos and then CoImmune, laid the foundation in 1997. Following this, CureVac was founded three years later. In 2008, BioNTech, Novartis, and Shire established their mRNA divisions, and Moderna was established in 2010.

mRNA serves as the key element in vaccine development, encoding specific antigens responsible for eliciting humoral and cell-mediated immunity and thus establishing a protective barrier against infections. However, its single-strand structure and susceptibility to environmental RNase digestion render mRNA highly unstable. To address this issue, essential stabilizing elements are employed, including 5′ capping with the m7GPPPN structure, a 3′ poly-A tail, and encapsulation in lipid nanoparticles (Figure 1). Optimizing mRNA vaccine design and antigen screening involves integrating various critical features. These include the incorporation of the Kozak sequence, efficient promoters, codon optimization, and UTRs to ensure robust antigen expression. Additionally, elements such as P2A, signal peptides, transmembrane domains, and flexible linkers play a crucial role in guaranteeing the correct folding and secretion of polyproteins. To enhance immune responses and enable multivalent targeting, polymer tricks are employed. Strategies such as incorporating sequences like FOLDON, which was used in BNT162b1 vaccines designed by Pfizer/BioNTech [2], or introducing disulfide bonds through cysteine and proline modifications (e.g., S2P, S6P, or SOSIP [3,4]), facilitate the production of homologous and heterologous trimers or tetramers. Furthermore, peptide sequence truncation, recombination, and amino acid substitutions are frequently employed for tailored structural modifications. Through these reasoned adjustments and manipulations, mRNA can be harnessed as a stable and functional conductor of immune cascade reactions.

Conventional vaccines typically utilize natural pathogen proteins, whole microbes that are inactivated or attenuated, or vectored recombinant proteins and organisms as immunogens to elicit immune responses and generate memory lymphocytes upon re-exposure to related epitopes or peptides. However, they face significant drawbacks when countering emerging outbreaks effectively in clinical practice. In contrast, mRNA vaccines represent a more advanced technology with distinct advantages. They can be rapidly and affordably manufactured, exhibiting high flexibility and adaptability to address emerging variants and outbreaks. mRNA vaccines achieve this by encoding any desired antigens or combinations through simple modifications, making them ideal for handling viruses with frequent mutations, such as influenza and coronaviruses. Moreover, mRNA vaccines can serve as both preventive and therapeutic measures, inducing robust humoral and CD8+ T cell responses simultaneously. For pathogens that can hide inside host cells and remain immunotolerant, such as *Mycobacterium tuberculosis*, *Brucella abortus,* and *B. melitensis*, the induction of CD8+ T cell responses becomes crucial for microbial clearance and long-term protection. CD8+ T cell responses are also essential for rapid recovery following infection. Compared to DNA vaccines, another advanced technique, mRNA vaccines offer faster and more efficient antigen expression, as mRNA can be directly translated in the cytoplasm, bypassing the nuclear transcription step required by DNA vaccines. Furthermore, mRNA vaccines and their delivery components possess the ability to stimulate long-term adaptive immune responses through a self-adjuvant effect, eliminating the need for additional ones [5]. In 2016, Swaminathan G. et al. showcased the application of empty lipid nanoparticles (LNPs) as adjuvants in a dengue virus protein vaccine [6]. The ionizable lipid in LNP acts as an immune stimulator. Combining empty LNPs with SARS-CoV-2 RBD proteins elicits a more robust immune response than protein vaccines with AddaVax adjuvants [7]. Protein vaccines utilizing empty LNPs as adjuvants prompt immune effects in both humoral and cellular realms, similar to mRNA-LNP vaccines, with the LNP’s ionizable lipid playing a pivotal role [7]. Administering LNPs to mice leads to substantial IL-6 production, fostering the generation of Tfh cells and GC B cells [7]. Moreover, double-stranded RNA (dsRNA) and the double strand formed by the secondary structures of mRNA could influence immune responses. Intracellular molecules TLR3 and MDA5 are indispensable for dsRNA recognition [8,9]. In mouse experiments with the BNT162b2 vaccine, TLR3-deficient individuals have minimized the neutralizing antibody level, while MDA5-deficient ones significantly reduced the frequency of antigen-specific CD8 T cell proliferation. The MDA5-IFN-α pathway notably enhances mRNA vaccine-induced cellular immunity [10]. Additionally, the potential application of mRNA vaccines in cancer immunotherapy highlights their capability to trigger robust humoral and cellular responses, further showcasing their versatility and efficacy.

Since the first use of vaccines for cowpox in 1796 [11], researchers and clinicians have utilized them to prevent various infectious diseases for over two centuries. Traditional vaccine approaches, including inactivated, attenuated, recombinant protein, and vectored vaccines, have effectively controlled at least thirty infectious diseases worldwide, with smallpox being eradicated through immunization. However, certain pathogens have managed to overcome and adapt to human immunity, posing challenges for conventional vaccines. They struggle to provide long-term protection and pathogen clearance against latency microbes and chronic viruses. These pathogens manipulate host immune responses by regulating cytokine secretion, macrophages, natural killer cells, and other lymphocytes’ maturation, or by integrating into the host genome and maintaining latency for extended periods. Additionally, for viruses like Dengue virus (DENV), Zika virus (ZIKV), and other *flaviviruses*, antibody-dependent enhancement hinders the efficacy of classical vaccines. In the context of the COVID-19 pandemic, the remarkable success of the SARS-CoV-2 mRNA vaccines, mRNA-1273 (Moderna) and BNT162b2 (Pfizer), which have received emergency use authorization (EUA) from the FDA [12,13], highlights the significant advancements in mRNA vaccines. Recognizing the advantages of mRNA technology, prominent biotechnology companies have already developed numerous mRNA vaccine pipelines, targeting pathogens such as Varicella Zoster virus (VZV), Respiratory syncytial virus (RSV), Human immunodeficiency virus (HIV), and influenza virus, underscoring its critical importance in the future of vaccine development. However, challenges do exist in the mRNA vaccine field, as the reality of the SARS-CoV-2 pandemic has proven that conventional SARS-CoV-2 mRNA vaccines cannot control and eliminate community transmission during the pandemic; confirmed infectious cases have even climbed up after successful vaccination [14], the neutralizing antibody titers declined rapidly [15], and side effects were also observed after individual inoculations. Although many reasons and details could be discussed and used to defend the mRNA vaccine from these cons, such as many people refusing to follow quarantine rules strictly, the mRNA vaccines were released long after the pandemic started, and as a consequence, millions of people were already infected, which makes it impossible to eliminate the transmission, the administration route is also inappropriate because it is unable to generate mucosal robust immunity and functional sIgA to protect people from nuzzle infection. Such challenges suggest that continuous efforts are still needed to be addressed in the development of mRNA vaccines for combating emerging infectious diseases. The mRNA vaccine design must consider and improve long-lasting humoral immunity, lower immunogenicity delivery system, and robust neutralizing antibody induction.

### 1.2. CircRNA Vaccine

Circular RNA (circRNA) represents a distinct class of naturally occurring or synthetic closed-loop RNA molecules lacking 5′ and 3′ ends. Its unique covalent circular structure confers higher pharmaceutical and biological stability compared to linear mRNA, protecting it from exonuclease degradation. Similar to mRNA, natural circRNA encompasses both non-coding and protein-coding components [16]. Current research on circRNA has unveiled several functional traits, including acting as a molecular sponge, regulating cell activity, and influencing protein translation. Moreover, circRNA’s association with specific human diseases endows it with the potential to serve as a novel disease biomarker and therapeutic target [17,18]. However, the precise physiological function of natural circRNA remains elusive. Nevertheless, synthetic circRNA has emerged as a promising avenue for the development of preventive and therapeutic vaccines, holding great promise for future medical applications.

Traditional linear mRNA translation relies on eukaryotic initiation factors and the 5′ cap structure. However, under stress conditions, cap-independent mechanisms mediated by Internal Ribosomal Entry Site (IRES) and m6A modifications can act as alternative pathways for eukaryotic RNA translation. CircRNA, lacking a 5′ cap, utilizes IRES and m6A RNA modifications for translation. The existence of IRES was first discovered by Macejack and Sarnow in 1991 when they found that poliovirus-infected host cells still expressed Immunoglobulin Heavy-Chain Binding Protein (BIP), suggesting IRES activity in the 5′ UTR of BIP mRNA [19]. In 1995, Chen and Sarnow demonstrated that synthetic circRNA containing EMCV IRES structures could express proteins in rabbit reticulocyte lysates [20]. Later, AbouHaidar and colleagues found a 220 nt long covalent closed circular RNA related to Rice Yellow Mottle Virus (RYMV) that was directly translated by eukaryotic ribosomes [21]. Subsequent studies identified other circRNAs, such as Circ-ZNF609, circ-FBXW7, circ-SHPRH, circPLNTexon2, circ-β-catenin, and circAKT3, that were translated into proteins via IRES in vivo [22,23,24,25,26,27]. Additionally, “IRES-like” elements or m6A RNA modifications can participate in the cap-independent translation of circular RNAs. A single m6A in the linear mRNA 5′ UTR can initiate translation by binding to eIF3 and recruiting the 43S complex. This M6A-driven translation requires initiation factors eIF4G2, m6A reader YTHDF3, methyltransferases METTL3/14, and demethylase FTO [28,29]. Moreover, Fan and his team identified IRES hexamer-like sequences enriched in endogenous circular RNAs, initiating translation through trans-acting factors binding to the IRES hexamer region [30]. While m6A represents a substitute mechanism for circular RNA translation, further confirmation is needed to elucidate the translation of endogenous circRNAs. As the field progresses, the role of IRES and m6A in circular RNA translation continues to be a subject of active research.

The widespread presence of circular RNA in vivo, along with its unique structural and functional properties, has generated a demand for efficient in vitro methods to prepare circular RNA, as illustrated in Figure 1. Currently, two primary approaches are employed for in vitro synthesis of circular RNA: chemical synthesis and enzymatic methods. However, most of these methods are better suited for small to medium-sized RNA [31]. Chemical synthesis of circular RNA ensures homogenization of the 3′ and 5′ ends, while enzymatically produced circular RNA often leads to reduced RNA cyclization yield due to terminal heterogeneity when the precursor RNA is synthesized by in vitro transcription [31,32,33]. Among the enzymatic approaches, the most commonly used methods for in vitro cyclization of long RNA involve the use of enzymatic ligases, such as T4 ligase. Meanwhile, ribozyme-based approach ribo-self-clipping intron Group I and Group II are the most popular in vitro circular RNA synthesis methods.

T4 ligase, derived from T4 phage, is commonly employed for enzymatic RNA ligations. For efficient ligation, the 5′ end of the linear RNA substrate should be a monophosphate nucleotide. If RNA is produced through an in vitro transcription (IVT) reaction using triphosphate nucleotide as the substrate, the phosphate group is removed through dephosphorylation, and a phosphate group is added to the 5′ end of the RNA via phosphorylation [31]. T4 ligase for RNA cyclization encompasses three types: (1) T4 DNA ligase, which catalyzes the repair of incisions in double-stranded substrates containing DNA. In this case, the two strands at the junction must be completely complementary and paired. However, its cyclization efficiency and the length of cycled RNA are limited [34,35]. (2) T4 RNA ligase 1, which acts solely on single-stranded substrates, making the reaction specificity relatively low [36]. This method involves designing splints with 10–20 nucleotides at both ends of the RNA molecule to achieve hybridization and complementation, generating circular RNA [31]. Sequences with complete complementary pairings typically exhibit the highest cyclization efficiency. However, this process requires several enzymes, and the turnover efficiency is low. (3) T4 RNA ligase 2, also used successfully for RNA cyclization, is compatible with both single-stranded and double-stranded RNA ligation [37]. It can achieve cyclization through complementary pairing within the molecule or by introducing exogenous splint sequences [38,39].

Group I intron: Puttaraju and Been demonstrated in 1992 that Group I intron is suitable for constructing circular RNA molecules [40]. The most widely used Group I introns, Anabaena pre-tRNALeu and T4td, are rearranged to facilitate cyclization: the 5′ end of the Group I intron is placed at the tail end of the exon, while its 3′ end is positioned at the head of the exon [40,41]. By adding exogenous GTP to attack the 5′ intron splice site, the 5′ intron is cleaved, and the hydroxyl group at the cleavage site can further attack the 3′ intron splice site, resulting in exon cyclization. Ford and Ares replicated this strategy in 1994 using a similar rearrangement of the introns of the T4 phage thymidylate synthase (td) gene, leading to the cyclization of the td exon. Since then, the rearranged T4 td and pre-tRNALeu genes have become the main templates for the Group I intron cyclization method, enabling the cyclization of various exon sequences inserted between the two halves of the Group I intron. Wesselhoeft et al. reported that circRNA production significantly increased when luciferin sequences were transferred from PIE constructs with td introns to PIE constructs with Anabaena pre-trNA introns, suggesting that cyclization efficiency varied between different Group I introns. Using the Anabaena skeleton, circRNAs up to 5000 nucleotides in length can be cyclized, while chemical and enzymatic linking strategies typically achieve cyclization of RNA molecules around 1000 nucleotides with a lower cyclization yield [42].

Group II intron: The Group II intron ribozyme self-splicing system operates similarly to the mechanisms of Group I intron and precursor mRNA introns. Mikheeva et al. (1997) demonstrated that rearranging Group II intron from yeast mitochondria generated a circular RNA without an exon sequence, with its ring formation connected by a 2′-5′ phosphodiester bond, while the ring formation of Group I intron is connected by a 3′-5′ phosphodiester bond. However, the exact in vitro mechanism by which this occurs remains unknown [43]. Recently, a novel coronavirus RNA [44] was designed using the tetanus Group II intron, but further investigation is needed to determine the mechanism details and overall applicability of this method.

### 1.3. saRNA Vaccine

Self-amplifying RNA (saRNA) vaccines are genetic vaccines that utilize modified RNA to induce immune responses against specific pathogens [45]. They have garnered significant attention for their potential to rapidly and effectively prevent infectious diseases, including viral infections like influenza and SARS-CoV-2 [46,47]. The key distinction between saRNA vaccines and conventional mRNA vaccines lies in their ability to amplify the RNA within cells, leading to increased protein production and potentially stronger immune responses, as depicted in Figure 1. This amplification feature is inspired by alpha-viruses, a group of positive-strand RNA viruses known for their efficient replication within host cells [48].

To create saRNA vaccines, scientists have harnessed the replicative machinery of alphaviruses and modified it to carry the desired genetic information for vaccine development. The replication elements typically used in saRNA vaccines are derived from alphaviruses like the Semliki Forest virus (SFV) [49,50] or the Venezuelan equine encephalitis virus (VEEV) [49,51]. These elements include non-structural proteins, such as the RNA-dependent RNA polymerase (RdRp), which play a vital role in replicating the RNA genome of alphaviruses. By incorporating these alphavirus-derived replication elements, the saRNA vaccine can undergo multiple rounds of replication and amplification within host cells, resulting in increased production of the desired viral protein(s). This approach offers several advantages, such as efficient and robust RNA replication leading to higher levels of viral protein expression within cells, enhancing the vaccine’s immune response. It is essential to emphasize that while saRNA vaccines are inspired by alphaviruses, they do not cause the same pathology or symptoms associated with natural alphavirus infections. The replication elements of alphaviruses are carefully modified and engineered to ensure safety and optimal vaccine performance. Researchers have leveraged their knowledge to design saRNA vaccines with improved stability, translation efficiency, and immunogenicity. Importantly, saRNA vaccines have the potential to elicit robust immune responses with lower doses of RNA, making them an attractive platform for rapid and scalable vaccine development against various infectious diseases. The saRNA technology comprises three key components [48]: (1) Antigen coding sequence: This segment of the RNA contains instructions to produce the desired viral protein(s) responsible for eliciting an immune response. Often, this sequence is derived from the target pathogen, such as a specific viral surface protein. (2) RNA replication elements: These elements enable saRNA to undergo self-replication and amplification within host cells. They typically include components from viruses known for efficient RNA replication, like non-structural proteins from positive-strand RNA viruses such as alphaviruses. (3) Additional modifications: saRNA vaccines may include specific modifications to enhance stability, improve translation efficiency, and reduce immune recognition, thereby contributing to a more potent immune response.

It is essential to acknowledge that saRNA vaccines are still a relatively new technology and have not been widely employed for human vaccination. However, they hold significant potential for preventing and controlling various infectious diseases, including viral infections. Ongoing research and clinical trials are evaluating their safety and efficacy [52].

## 2. RNA Vaccines for Infectious Diseases

### 2.1. Coronavirus

As of July 2023, the SARS-CoV-2 pandemic has resulted in over 768 million confirmed cases and more than 6.9 million deaths [53]. SARS-CoV-2 is an enveloped single-stranded RNA virus with a high mutation rate, leading to the emergence of several variants of concern (VOCs) from different lineages. Notable VOCs include Alpha, Beta, Gamma, and the more recent Omicron variant, identified in November 2021, consisting of three sister lineages (BA.1, BA.2, and BA.3) [54], shown in Figure 2. These variants exhibit varying epidemiological characteristics, such as contagiousness, severity, and mortality. Traditional vaccine strategies cannot outpace the spread and evolution speed of the SARS-CoV-2 virus for a couple of reasons. Firstly, the preclinical study is exceptionally time-consuming, and usually takes decades before entering the clinic [55], since either molecular cloning, viral vector attenuation, mutation variant construction, viral passaging, and inactivation, or large-scale protein synthesis and purification require lots of experience and time. Secondly, traditional vaccines may be unable to trigger robust immunity and induce cross-protection to fight against the pathogen with high variability. Lastly, the relatively low scalability of traditional vaccines is another limitation when countering emerging outbreaks. On the contrary, the mRNA vaccine could offer a lightning-speed response to emerging infectious diseases, as proven in the SARS-CoV-2 pandemic. Due to its flexibility, by simply modifying and designing the mRNA sequence, accommodated RNA vaccines against different variants could be implemented in practice in a week by finishing in vitro transcription, encapsulation, and related quality control. By adding immuno-RNA elements, such as FOLDON, SOSIP, MITD, etc., the RNA vaccine could be more specific and motivated, properly inducing humoral and cell immunity from various signal pathways on purpose. On the other hand, an RNA vaccine would not consider the pathogen proteins’ expression and fold as issues since mRNA is directly delivered into eukaryotic cells, and mother nature will generate the native protein using cell machinery, no matter how complicated the variant is, unlike some recombinant protein vaccines expressed in prokaryotes with inappropriate glycosylation and folding.

Significant efforts have been devoted to developing prophylactic mRNA vaccines against SARS-CoV-2 over the past 3.5 years. Corbett et al. pioneered the design of a prototype pathogen-based non-replicating mRNA vaccine, mRNA-1273, encoding the SARS-CoV-2 spike (S) protein with specific proline substitutions [56]. Only after 34 days, when Chinese scientists revealed the genome sequence of the mRNA, Moderna submitted its first doses of the mRNA SARS-CoV-2 vaccine to the US National Institute of Health (NIH), and the clinical trial was conducted 21 days later. This unprecedented development speed makes mRNA a powerful weapon in fighting the pandemic. mRNA-1273 has demonstrated efficacy in eliciting potent neutralizing antibodies and T-cell immunity in preclinical studies and clinical trials [12]. Similarly, BNT162b2, designed with a similar rationale as mRNA-1273 [57], has shown exceptional efficacy against the original SARS-CoV-2 strain [58]. However, the effectiveness of these vaccines has shown signs of waning over time, with declining neutralizing IgG antibodies and the emergence of new variants. Studies suggest that booster doses of mRNA-1273 or BNT162b2 can enhance neutralizing antibodies against Omicron and the Delta variant [59,60]. In response to the evolving pandemic, Omicron-adapted mRNA vaccines have been developed. These vaccines, such as Omicron-adapted BNT162b2 and ZSVG-02-O by Sinopharm, exhibit a remarkable boost in neutralizing antibodies. ZSVG-02-O shows cross-activity against multiple strains, including the wild type, Delta, and BA.1 VOCs, offering critical advantages in combating future variants [61]. Pharmaceutical companies, including Spikevax, Sinopharm, Pfizer, and Moderna, have also explored bivalent vaccine strategies targeting two variants simultaneously, often including the wild type and Omicron variants [62]. These developments reflect the continuous efforts to combat the evolving SARS-CoV-2 virus and underscore the importance of mRNA vaccines in the fight against infectious diseases.

Both saRNA and circRNA technologies have significantly contributed to the advancement of SARS-CoV-2 vaccine development, addressing the severity of the pandemic. McKay et al. pioneered the development of a saRNA-based vaccine that encodes a stabilized SARS-CoV-2 spike protein in its pre-fusion conformation. Pre-clinical studies in mouse models have demonstrated its ability to induce robust Th1-mediated immunity, potent neutralizing antibodies against SARS-CoV-2, and notable cytokine responses [63]. Encouraging clinical results show that 80% of sera samples from vaccine recipients generated neutralizing antibodies after the second dose, and all participants who tested positive for SARS-CoV-2 exhibited a significant increase in spike-specific IgG levels [64]. Importantly, saRNA vaccines can achieve desirable immunogenicity at relatively lower doses compared to conventional mRNA vaccines, owing to their self-amplification property. This suggests that saRNA vaccines may offer a more efficient immunization approach at ultra-low doses and potentially shorten the immunization interval considerably. On the other hand, circRNA has been utilized as a vaccine approach to combat SARS-CoV-2. Researchers, led by Qu and colleagues, developed a circRNA encoding the trimeric Receptor Binding Domain (RBD) for both Delta and Omicron variants. The design employed the group I intron autocatalysis strategy with an IRES-SP-RBD-FOLDON construct, where the IRES element, followed by the human tissue plasminogen activator (tPA) signal peptide, preceded the RBD coding sequence. The addition of FOLDON to the RBD C-terminus facilitated RBD trimerization. The results revealed that circRNA induced robust Th1-mediated immune responses, resulting in higher levels of neutralizing antibodies and broader cross-activity against Delta and Omicron variants. This makes circRNA a promising candidate for a broad-spectrum vaccine approach. In conclusion, both saRNA and circRNA technologies have demonstrated their potential in advancing SARS-CoV-2 vaccine development, offering sophisticated and promising strategies to address the challenges posed by the pandemic.

### 2.2. Flavivirus

Flaviviridae is a family of viruses primarily transmitted by blood-feeding arthropod vectors, causing a variety of diseases with pandemic potential [65], including Zika virus (ZIKV) [66], Dengue virus (DENV) [67], Tick-borne encephalitis virus (TBEV) [68], West Nile virus (WNV), Japanese encephalitis virus (JEV), and Yellow Fever virus (YFV) [69]. The spherical flavivirus particles are approximately 40–60 nm in diameter and consist of a 30-nm core region and a lipoprotein membrane. The viral nucleocapsid contains single-stranded sense genomic RNA encapsulated by bi-layer lipid membrane glycoproteins (E and M proteins). mRNA vaccines against flaviviruses have been developed since 2013, with promising results. Moderna developed mRNA-1893, encoding the prM and E proteins of the RIO-U1 strain of ZIKV, and the vaccine showed robust ZIKV-specific neutralizing antibody responses in phase 1 clinical trials. The vaccine provided effective serum-neutralizing antibodies and long-lasting immunity, persisting up to month 13 [70]. Furthermore, it offered complete protection against the ZIKV challenge in nonhuman primates [71]. However, another Moderna mRNA vaccine, mRNA-1325, induced 20 times lower neutralizing antibody titers compared to mRNA-1893. Pardi et al. demonstrated that a single low-dose immunization of mRNA-LNP encoding ZIKV membrane and envelope (prM-E) glycoproteins protected mice and nonhuman primates from ZIKV infection [72]. A dose of 30 μg of ZIKV mRNA-LNP vaccine conferred protection against the ZIKV challenge at two weeks or five months after vaccination, with no viremia detected in vaccinated mice and rhesus monkeys after the ZIKV challenge. Roth et al. reported that immunization of HLA class I transgenic mice with deny-NS, an mRNA vaccine consisting of immunogenic NS3, NS4B, and NS5 proteins, induced robust CD8+ T cell immune responses and significant protection against DENV1 challenge [73]. Furthermore, modified mRNA vaccines expressing prM and E proteins of the JEV P3 strain produced effective neutralizing antibodies and CD8+ T lymphocyte-mediated immune responses in mice. This vaccine provided protection against the lethal JEV challenge and reduced neuroinflammation [74]. Additionally, Lex G. et al. demonstrated protection of mice from lethal YF virus infection through passive injection of serum or spleen cells from mice immunized with mRNA vaccines of YF virus membrane and envelope proteins or nonstructural protein 1 (NS1). Macaques immunized with mRNA vaccines sustained high humoral and cellular immune responses for at least five months after the second vaccination dose. Overall, mRNA vaccines have shown promising immunogenicity and protection against flaviviruses, making them potential candidates for combating these infectious diseases [75].

### 2.3. Influenza Virus

Seasonal epidemics caused by influenza A and B viruses lead to millions of severe cases and hundreds of thousands of respiratory deaths annually [76]. Currently, flu shots are available, targeting two glycoproteins, hemagglutinin (HA) and neuraminidase (NA), of four WHO-alerted flu viruses [77]. The success of the COVID-19 mRNA vaccine has inspired the development of a new generation of influenza mRNA vaccines. Multiple influenza mRNA vaccines have been released since 2019 [78]. Moderna’s quadrivalent seasonal influenza vaccine candidate (mRNA-1010) and Pfizer’s quadrivalent mRNA flu vaccine have shown promising results in Phase 3 studies. The mRNA vaccines exhibited robust immune responses against all four circulating strains, indicating the advantage of the mRNA platform [79]. In another case, Pfizer’s single-dose flu vaccine candidate exhibited significantly enhanced CD4+ and CD8+ T cell responses against all four circulating strains in 65+ adults [80]. However, current seasonal influenza vaccines offer only 40% to 60% effectiveness, and sometimes as low as 10%, due to the antigenic drift and shift of the influenza virus [81]. To address this limitation, the focus is now on developing universal flu vaccines that provide broader and longer-lasting protection. Moderna’s mRNA-1011/1012 and mRNA-1020/1030 [82], as well as Pfizer’s self-amplifying RNA (saRNA) vaccine pipeline [83], aim to provide near-universal protection against selected strains of the virus. Furthermore, studies have explored multivalent mRNA vaccines encoding conserved influenza virus antigens, inducing antigen-specific immune responses and broad protection. A notable development is the 20-HA mRNA-LNP vaccine encoding HA from all known influenza A virus subtypes and influenza B virus lineages, which demonstrated protection against matched and mismatched viral strains in mice and ferrets [84]. As the quest for a universal influenza vaccine continues, mRNA technology offers a promising strategy to improve strain matching, induce broader immune responses, and generate long-lasting antibodies [85,86]. The advancement of mRNA vaccines represents a significant milestone in the pursuit of a universal influenza vaccine.

### 2.4. Respiratory Syncytial Virus (RSV)

RSV (Respiratory Syncytial Virus) is a negative-sense, single-stranded RNA virus mainly transmitted through direct or indirect contact. It has two main serotypes, RSV-A and RSV-B, that circulate primarily during the winter and spring. RSV can cause upper respiratory tract infections (URTI) or lower respiratory tract infections (LRTI), with children and elderly patients being particularly vulnerable to severe infections. Notably, RSV is one of the most prevalent viral pathogens causing acute lower respiratory tract infections (ALRTI) in children under the age of 5 globally, and it remains the primary cause of hospitalization for infants with viral respiratory tract infections. This places a significant burden on public health and healthcare systems [87,88,89,90]. The incidence and healthcare costs associated with RSV infection vary across countries with different levels of development. Low- and middle-income countries experience a higher incidence of RSV lower respiratory tract infections compared to high-income countries. However, the hospitalization rate in low- and middle-income countries is lower, likely due to limited medical resources leading to inadequate treatment for severely ill children. RSV infection has a higher mortality rate compared to other causes of lower respiratory tract infections, making it a major concern for healthcare services. Among children under 5, the number of RSV-related emergency visits and hospitalizations is significantly higher than those caused by the influenza virus. In China, the incidence of acute lower respiratory tract infections caused by RSV is estimated to be approximately 31/1000, accounting for 18.7% of childhood acute lower respiratory tract infections. RSV also poses a significant threat to the health of the elderly population, with the aging population leading to increased incidence and mortality rates of respiratory tract infections, including RSV. Individuals with compromised immune function and the elderly are at higher risk of severe illness and potential life-threatening consequences due to RSV. Moreover, RSV can worsen underlying conditions such as COPD, asthma, and chronic heart failure, leading to pneumonia, hospitalization, and mortality. Given RSV’s significant economic burden on global healthcare services, preventive vaccines against RSV hold great promise. Developing effective RSV vaccines can help alleviate the impact of RSV-related infections, particularly in vulnerable populations, and improve overall public health outcomes.

The respiratory syncytial virus (RSV) genome encodes 11 proteins, with the surface proteins, Glycoprotein (G) and Fusion protein (F), playing pivotal roles in the virus attachment and fusion processes. These proteins are crucial antigenic sites for generating neutralizing antibodies and act as primary targets for inducing host immune responses and antiviral defenses [91]. Among these proteins, the F protein exhibits high conservation across RSV subtypes, making it a promising target for cross-protective neutralizing antibodies. In vaccine design and antibody development, the pre-fusion (pre-F) conformation of the F protein is of particular significance, as it plays a more critical role in neutralization compared to the post-fusion (post-F) conformation. The pre-F conformation contains a unique antigenic site known as Ø, which triggers a robust neutralizing antibody response more effectively than other antigenic sites. Studies have demonstrated that during natural RSV infection, neutralizing antibodies are primarily induced by the Ø site in the pre-F form [92]. Therefore, stabilizing the Ø antigenic site is vital for the development of effective RSV vaccines. Notably, designs such as DS-Cav1 and DS2, developed by McLellan, Joyce, et al., have successfully stabilized the RSV F protein in the pre-fusion conformation, significantly advancing RSV vaccine research [92,93,94,95,96]. These breakthroughs hold promise for improving preventive measures against RSV infection and represent significant advancements in the field of RSV vaccine development.

In clinical practice, two available neutralizing antibodies for the respiratory syncytial virus (RSV) are palivizumab and nirsevimab (Table 1) [97,98], both targeting the pre-F conformation. Palivizumab requires multiple injections, making it more expensive and recommended primarily for high-risk infants and children. In contrast, nirsevimab, developed by AstraZeneca/Sanofi, is a long-acting injectable neutralizing antibody specifically targeting the RSV preF protein, administered as a single dose. Nirsevimab has demonstrated remarkable efficacy, with a 74.5% reduction in RSV lower respiratory tract infection (LRTI) incidence, and obtained marketing approval from the European Union in November 2022. Active protection against RSV infection is essential, and currently, more than 30 vaccine candidates have entered clinical trial stages. Significant progress has been made by companies such as AstraZeneca/Sanofi, GSK, Pfizer, Johnson & Johnson, Bavarian Nordic, and Moderna in RSV vaccine development [99,100]. Notably, GSK’s RSVPreF3 OA (Arexvy, showed in Table 1), a recombinant subunit pre-fusion RSV F glycoprotein antigen vaccine for the elderly, has shown overall efficacy of 82.6% against RSV lower respiratory tract disease (RSV-LRTD) and received FDA approval in May 2023. GSK is also conducting Phase III studies for a maternal immunization vaccine utilizing a recombinant subunit pre-fusion RSV F glycoprotein antigen, with results expected in early 2024. Pfizer’s bivalent pre-fusion F subunit vaccine, Abrysvo (Table 1), developed explicitly for maternal immunization, has demonstrated over 80% protection in newborns and received FDA approval in May 2023 for preventing RSV-associated lower respiratory tract disease (LRTD) in individuals aged 60 and above. However, GSK’s RSVPreF3 OA has shown superior efficacy to Abrysvo, likely attributed to the AS01 adjuvant used [101]. Moderna’s RSV mRNA vaccine, mRNA-1345 (Table 1), encoding the RSV preF protein, demonstrated good tolerability and efficacy in Phase III clinical trials, with an efficacy of 83.7% in preventing two or more RSV-associated lower respiratory tract diseases (RSV-LRTD) in the elderly population. Most RSV preventive drugs target the RSV pre-F conformation, although some vaccines utilize other antigens. Advaccine (Table 1), for example, has developed subunit vaccines targeting the G protein antigen for the elderly and children, with ongoing Phase II clinical trials in Australia. Another notable vaccine candidate is MVA-BN RSV (Table 1), which encompasses five RSV antigens and was granted breakthrough therapy designation by the FDA in February 2022, recognizing its potential as a significant advancement in RSV treatment. These vaccine developments show promising progress in the fight against RSV infection.

## 3. RNA Vaccine Delivery System

### 3.1. Lipid Nanoparticles (LNP)

RNA vaccines have emerged as a highly sophisticated and effective vaccine technology, but they still face two key challenges that limit their effectiveness. Firstly, negatively charged RNA has difficulty crossing the cell membrane structure, and secondly, RNA, especially mRNA, is intrinsically unstable and susceptible to degradation by ribonucleases in the circulation. To address these challenges, drug delivery systems have been developed to shield and protect the RNA. One of the most successful RNA delivery approaches is the lipid nanoparticle (LNP), which has achieved great success during the SARS-CoV-2 pandemic. LNP is composed of four main components: an ionizable or cationic lipid to neutralize the RNA’s negative charge, a zwitterionic lipid that has minimal electrostatic interaction with blood cells’ membranes, cholesterol to stabilize the LNP’s membrane, and PEG, which affects particle size and colloidal stability (as shown in Figure 3a) [102]. This four-component design offers several advantages, including scalable production, efficient RNA encapsulation, and transfection of antigen-presenting cells. The most notable examples of LNP-based delivery systems are the SARS-CoV-2 prophylactic vaccines mRNA-1273 and BNT162b2, both utilizing the four-component formulation with minor structural adjustments and lipid compositions [103].

Lipid nanoparticle (LNP) is currently the most widely used drug delivery system for RNA vaccines. However, there are significant opportunities for improvement. First, thermostability remains a challenge as these vaccines require extremely low transportation and storage temperatures. Second, the inflammatory nature of LNP components can lead to undesirable side effects or even fatalities with different administration routes [104]. Recent research focuses on developing lyophilized LNP formulations, exploring better cryo-preservative or lipid alternatives, and investigating new-generation delivery systems. These efforts aim to enhance the technology and address its limitations effectively.

### 3.2. Extracellular Vesicles (EVs)

Extracellular vesicles (EVs) have emerged as a promising natural RNA delivery carrier due to their superior biocompatibility. These nanosized structures, enclosed by phospholipid bilayers, are released by mammalian and bacterial cells and contain proteins, lipids, and nucleic acids. EVs play a crucial role in cell-to-cell communication by delivering functional molecules. Mammalian EVs can be categorized into exosomes, microvesicles, and apoptotic bodies based on their biogenesis routes [105]. Exosomes, originating from vesicles inside multivesicular bodies (MVBs) and released during the fusion of MVBs with the plasma membrane (Figure 3b), are particularly promising as mRNA drug delivery vehicles due to their safety profile. However, efficient mRNA encapsulation inside exosomes remains a critical challenge in their use as a delivery system. Tsai et al. explored mRNA encapsulation using cationic lipids mixed with Antares2 mRNA, which was then incorporated into exosomes or attached to the phospholipid bilayers when combined with exosomes. The study revealed that Antares2 mRNA-loaded exosomes exhibited higher mRNA expression than LNPs in vitro and demonstrated expression in mice models [106]. Moreover, the researchers loaded exosomes with mRNA encoding the SARS-CoV-2 fusion protein LSNME, which translated into nucleocapsid protein (N), membrane protein (M), and envelope protein (E). This mRNA-loaded exosome induced a robust humoral and cellular response in animal experiments without significant adverse effects [106]. These findings demonstrate the excellent biocompatibility and potential of exosomes as an RNA delivery system, although the functional mRNA cargoes were inserted into the phospholipid bilayers rather than the exosome lumen, indicating further exploration is warranted.

Genetic engineering presents a promising approach for loading RNA cargo into the lumen of extracellular vesicles (EVs). Hung et al. introduced the ‘TAMEL approach (targeted and modular EV loading approach)’ to achieve dynamic mRNA loading. This method involves fusing the RNA-binding protein MS2 bacteriophage coat protein to the surface EV-enriched protein (EEP) LAMP2b, enabling high loading efficiency. Their study revealed that smaller RNAs (500 kb) had higher loading efficiency compared to longer RNAs (~1.8 kb). Additionally, the choice of EEP significantly impacted the loading efficiency, with VSVG showing a 42-fold enrichment of cargo RNA in vesicles compared to Hspa8, which exhibited minimal enrichment [107]. Another recent study used two constructs to facilitate dynamic mRNA loading into exosomes. One construct expressed a fusion protein of the RNA binding domain ‘PUFe’ and the EEP ‘CD63’, while the other expressed the mRNA of interest. During exosome biogenesis, the mRNA cargo was selectively packaged into the exosome lumen. In vitro experiments demonstrated that Nanoluc mRNA could be selectively packaged inside exosomes using this approach, and the Nanoluc mRNA was functionally delivered and translated. Notably, the researchers enhanced mRNA stability by introducing PABPc, a protein that protects mRNA from 3′UTR-directed degradation via the nonsense-mediated decay pathway, by binding to the poly-A tail. Co-expressing PABPc in the donor cells substantially improved loading efficiency. By decorating the exosomes with Vesicular Stomatitis Virus g (VSVg), the fusogenic escape efficiency increased, allowing successful mRNA delivery at a much lower dose than LNP in mice models [108]. These innovative genetic engineering strategies hold great promise for improving the efficiency and functionality of RNA delivery using EVs.

Bacterial extracellular vesicles (EVs) display a diverse range of structures and compositions. Among them, outer membrane vesicles (OMVs) are shed from both Gram-negative and positive bacterial organisms (Figure 3c). OMVs can selectively package specific cargoes such as virulence factors, lipids, and DNA, making them crucial for functional communications between bacteria-bacteria, bacteria-host, and bacterial survival [109]. Over the past decade, OMVs have emerged as a promising mRNA delivery system due to their intrinsic immunogenicity, characterized by the display of ‘pathogen-associated molecular patterns (PAMP)’ on their surface. Moreover, OMVs can be manufactured on a large scale through simple fermentation. Researchers have explored genetic engineering to load mRNA onto the surface of OMVs effectively. For instance, Li et al. utilized an RNA-binding protein (L7Ae) fused to the anchor protein cytolysin A, which is abundantly expressed on the surface of OMVs, to specifically decorate OMVs with mRNA of interest. The mRNA was labeled with a binding sequence (e.g., box C/D sequence) to achieve strong and specific conjugation between the RNA-binding protein and the mRNA. To enhance mRNA translation productivity, listeriolysin O (LL) was further added to the surface of OMVs to facilitate endosome escape. This engineered OMV-LL-mRNA was successfully employed as a colon cancer therapeutic vaccine, inducing 37.5% complete regression in experimental models [110]. This research demonstrates the promising potential of OMVs as an efficient mRNA delivery platform for therapeutic applications.

### 3.3. Other Delivery Systems (Protamine and Hydrogel)

Protamine is a naturally occurring protein derived from salmon spermatozoa, containing a significant number of positively charged L-arginine amino acids, making it ideal for condensing and stabilizing negatively charged mRNA [111]. Studies have shown that protamine-mRNA complexes significantly enhance mRNA transfection efficiency, resulting in robust antigen-specific CD4+ T cell, CD8+ T cell, and B cell responses in mice models [112]. To further improve the transfection efficiency of the protamine-based delivery system, researchers have incorporated endosome membrane destabilizing agents, such as poly(acrylic acid) polymers [111]. Hydrogels are soft, water-swollen, biodegradable, three-dimensional structures that can efficiently trap mRNA within their network (Figure 3d). Hydrogels offer several advantages as RNA delivery carriers. Firstly, they provide effective protection against the enzymatic degradation of mRNA. Secondly, nano-gel engineering allows for targeted delivery of mRNA to specific cells or tissues. Thirdly, hydrogels can achieve pulsatile or prolonged release, potentially enabling single-shot immunization, which could alleviate the need for multiple vaccine administrations [113]. Current mRNA vaccines, such as those against SARS-CoV-2, VZV, RSV, and influenza, typically require multiple injections. Replacing lipid nanoparticle (LNP) delivery systems with controlled-release hydrogels have the potential to streamline vaccination processes, providing a more convenient and efficient immunization strategy while reducing the burden on healthcare systems [113].

## 4. Discussion

RNA vaccines have emerged as a powerful tool in the fight against the SARS-CoV-2 pandemic, demonstrating significant advantages such as large-scale manufacturing capacity and reasonable cost, making them accessible to developing and underdeveloped countries, and creating a positive global impact. Different types of RNA, including conventional mRNA, self-amplifying mRNA, and circular RNA, offer distinct advantages and challenges. Conventional mRNA is safe for administration and easily manufactured via in vitro transcription (IVT); however, it may not always elicit the desired immune response, necessitating the use of an effective delivery system. Self-amplifying mRNA addresses this limitation by enabling sufficient antigen expression through its self-amplifying process, but the use of alphavirus elements may raise safety concerns and induce unwanted immune responses. Circular RNA shows promise due to its inherent stability, resulting in a longer RNA half-life and prolonged antigen expression without requiring a delivery system. However, its complex manufacturing process and limited RNA circulation pose challenges. Additionally, in some circumstances, RNA vaccines may be difficult to implement, as in resource-limited or low-income areas, poverty creates cost barriers for those regions using this advanced technology. RNA vaccines can be more expensive to produce and distribute if they require lyophilization or cold-chain transport to be stabilized. Despite these challenges, mRNA vaccines have achieved significant milestones not only in the SARS-CoV-2 pandemic but also in combating other pathogens, as evidenced by promising preclinical and clinical data for mRNA vaccines targeting flaviviruses, VZV, RSV, and influenza. Furthermore, ongoing efforts are directed toward developing mRNA vaccines for obstinate infectious diseases such as ‘*Mycobacterium tuberculosis*’, ‘HIV’, and ‘*Malaria*’. To further advance the field, attention should be directed towards addressing potential issues in manufacturing and RNA delivery systems, including concerns related to the toxicity of lipid components, and the need for low-temperature storage and transportation. With continued research and investment in this thriving area, these challenges are likely to be overcome in the near future.

## Figures and Tables

**Figure 1 viruses-15-01760-f001:**
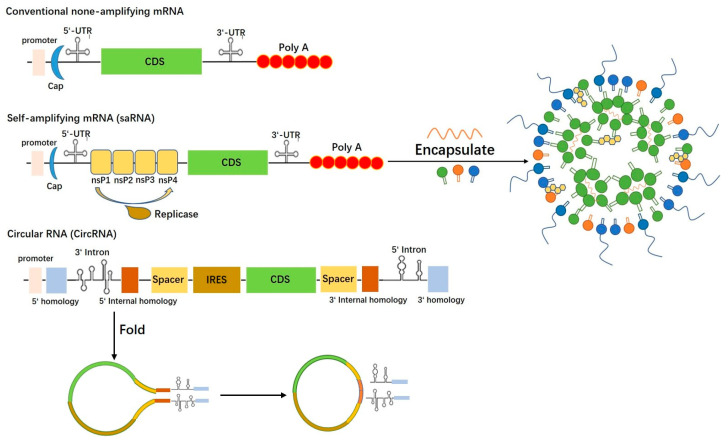
Schematic of conventional mRNA, saRNA, and CircRNA structures and encapsulation.

**Figure 2 viruses-15-01760-f002:**
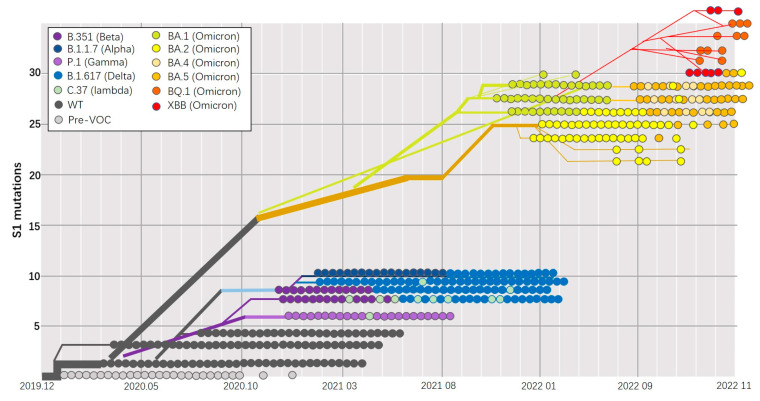
SARS-CoV2 lineage prevalence and mutation accumulation in spike S1 protein. This figure is modified and redrawn from “The evolution of SARS-CoV-2” [54].

**Figure 3 viruses-15-01760-f003:**
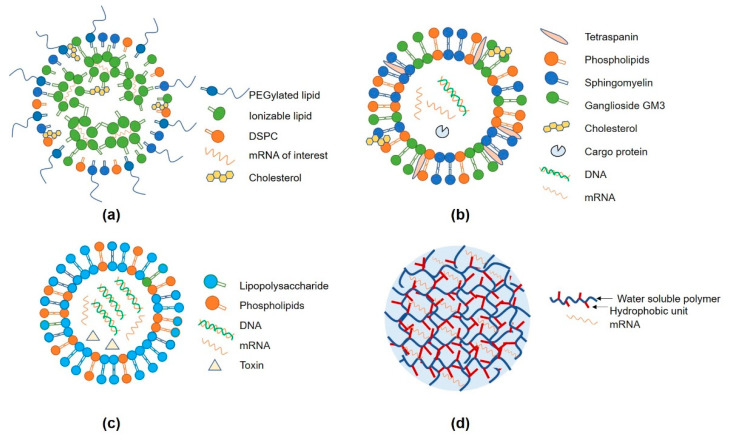
Structures and components schematic of four different drug delivery systems. (**a**) Structure of lipid nanoparticles (LNP) and its four main components, including PEGylated lipid, ionizable lipid, DSPC, and cholesterol; (**b**) Structure of the exosome. The exosomal lipids include phospholipids, sphingomyelin, ganglioside GM and cholesterol. The most abundant exosomal protein is tetraspanin, including CD9, CD63, and CD81; (**c**) Structure of outer membrane vesicle (OMV) and its main components, the main lipid types of OMV are lipopolysaccharide and phospholipids; (**d**) Structure of hydrogel and its main component, which is a water-soluble polymer with a hydrophobic unit.

**Table 1 viruses-15-01760-t001:** Worldwide RSV vaccine under development.

Name	Developer	Vaccine Type	Participants	Clinical Trial Status	Year of Approval
Beyfortus (Nirsevimab)	AstraZeneca/Sanofi	Anti-preF mAb (Immune-prophylaxis)	Infants (0 Days to 12 Months)	Phase 3 (NCT03979313): Attended RSV-LRTI through 150 days after the injection: 74.5%	2022
mRNA-1345	Moderna	RSV preF (Nucleic acid)	Elderly	Phase 3 (NCT05127434): RSV-LRTD with ≥2 symptoms: 83.7%; RSV-LRTD with ≥3 symptoms: 82.4%	
mRNA-1345	Moderna	RSV preF (Nucleic acid)	Infants (5–24 Months)	Phase 1 (NCT05743881)	
Arexvy (RSVPreF3)	GSK	RSV preF (Protein-based particle)	Elderly (≥60 years)	Phase 3 (NCT04886596) (AReSVi 006): RSV-LRTD with ≥1 symptoms: 82.6%; RSV-LRTD with ≥2 symptoms: 94.1%	2023
Abrysvo (RSV preF)	Pfizer	RSV preF A and RSV preF B (Protein-based particle)	Elderly (60–80 years old)	Phase 3 (NCT05035212): RSV-LRTD with ≥2 symptoms: 66.7%; RSV-LRTD with ≥3 symptoms: 85.7%	2023
Abrysvo (RSV preF)	Pfizer	RSV preF A and RSV preF B (Protein-based particle)	Maternal Immunization	Phase 3 (NCT04424316): 90-day-old infants with severe MA-LRTI: 81.8% 180-day-old infants with severe MA-LRTI: 69.4%	
BARS13	Advaccine	G protein (Protein-based particle)	Elderly (60–80 years old)	Phase 2 (NCT04681833)	
MVA-BN RSV	Bavarian Nordic	F, N, M2-1, G (subtype A), G (subtype B) (Recombinant vector)	Elderly (≥60 years)	Phase 3 (NCT05238025)	

## Data Availability

Not applicable.

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
