# Peer review of "Vaccines’ New Era-RNA Vaccine"

_viruses, 2023, doi:10.3390/v15081760_

Round 1

Reviewer 1 Report

This review provides an informative overview of RNA vaccines, emphasizing their potential in revolutionizing vaccine development, included conventional RNA vaccine, circlRNA vaccine and saRNA vaccine. It highlights the success of mRNA vaccines against Coronavirus, flavivrus, influenza virus and RSV as a pivotal moment in demonstrating the capabilities of RNA-based vaccines. . The advantages of RNA vaccines, such as high efficacy, adaptability, rapid manufacturing, and the ability to induce both humoral and cellular immunity, are well-presented. They also discussed several delivery system, delve the advantage and shortage. Furthermore, they acknowledge some challenges that need to be addressed, including the technology's immaturity, research expenses, limited duration of antibody response, mRNA instability, and production of double-stranded RNA as a side product. It also mentions the potential of circular RNA as a self-adjuvant and discusses its role in enhancing RNA vaccines' effectiveness. Overall, the review provides a comprehensive and balanced overview of the current state of RNA vaccines for infectious diseases. It covers their potential advantages and challenges, thereby offering valuable insights into the future of RNA vaccine development and utilization in combating various diseases. No issue was detected here.

Author Response

Question: This review provides an informative overview of RNA vaccines, emphasizing their potential in revolutionizing vaccine development, included conventional RNA vaccine, circlRNA vaccine and saRNA vaccine. It highlights the success of mRNA vaccines against Coronavirus, flavivrus, influenza virus and RSV as a pivotal moment in demonstrating the capabilities of RNA-based vaccines. The advantages of RNA vaccines, such as high efficacy, adaptability, rapid manufacturing, and the ability to induce both humoral and cellular immunity, are well-presented. They also discussed several delivery system, delve the advantage and shortage. Furthermore, they acknowledge some challenges that need to be addressed, including the technology's immaturity, research expenses, limited duration of antibody response, mRNA instability, and production of double-stranded RNA as a side product. It also mentions the potential of circular RNA as a self-adjuvant and discusses its role in enhancing RNA vaccines' effectiveness. Overall, the review provides a comprehensive and balanced overview of the current state of RNA vaccines for infectious diseases. It covers their potential advantages and challenges, thereby offering valuable insights into the future of RNA vaccine development and utilization in combating various diseases. No issue was detected here.

Answer: Thank you for the reading, efforts and very positive comments.

Reviewer 2 Report

General comment

The paper “Vaccines’ new era-RNA vaccineis an interesting and well-written article about RNA vaccines that provides a solid overview of the different types of RNA vaccines and highlights some of their potential advantages and challenges. However, there are certain aspects that warrant a critical analysis:

Major comments

The review appears to be overly optimistic about the potential of RNA vaccines without adequately addressing the limitations and uncertainties associated with this technology. It is essential to acknowledge that the long-term safety and efficacy of these vaccines require further monitoring and research. RNA vaccines, especially saRNA and circRNA vaccines, are relatively new and have not undergone extensive testing in large-scale clinical trials, which leaves room for uncertainties.

The review mentions the "self-adjuvant effect" of circular RNA or mRNA lipid nanoparticles, but does not elaborate enough on the potential risks and safety concerns with these adjuvants. Adjuvants can play a key role in enhancing the immunogenicity of the vaccine, but they can also cause unwanted side effects or adverse reactions in some people. A more in-depth discussion of the safety of adjuvants would provide a more balanced perspective.

Although mRNA vaccines have shown promising results in the fight against the COVID-19 pandemic, the reality is that they have not been able to definitively control and eliminate community transmission. This aspect should be noted in a section of Introduction to the article.

Specific comments

1) Write an “Introduction” to the text, coment the main sources of information  at the end specify the objective of this review.

2) The authors should further explain that developing and testing vaccines against new variants can be a complex and time-consuming process, and better explain how RNA vaccine technology can effectively respond to these challenges.

3) The review not explore the potential cost barriers that could limit access to these vaccines, particularly in resource-limited settings. RNA vaccines can be more expensive to produce and distribute, which might affect their availability in low-income countries and raise equity concerns.

4) The review provides a comprehensive overview of RNA vaccines and their potential, but it could benefit from a more critical approach.

Author Response

General comment

Question: The paper “Vaccines’ new era-RNA vaccine” is an interesting and well-written article about RNA vaccines that provides a solid overview of the different types of RNA vaccines and highlights some of their potential advantages and challenges. However, there are certain aspects that warrant a critical analysis: 

Answer: We followed your comments and did a more critical discussion on various aspects in this paper. We did all the corrections showing below with the suggestions you mentioned. Thank you for your efforts and professional input.

Major comments

Question: The review appears to be overly optimistic about the potential of RNA vaccines without adequately addressing the limitations and uncertainties associated with this technology. It is essential to acknowledge that the long-term safety and efficacy of these vaccines require further monitoring and research. RNA vaccines, especially saRNA and circRNA vaccines, are relatively new and have not undergone extensive testing in large-scale clinical trials, which leaves room for uncertainties.

Answer: We totally agree on “long-term safety and efficacy of these vaccines require further monitoring and research. RNA vaccines, especially saRNA and circRNA vaccines, are relatively new and have not undergone extensive testing in large-scale clinical trials, which leaves room for uncertainties.” More discussion on “long term safety and efficacy” has been added and more cons on RNA vaccines have been elucidated in the “conclusion and discussion” section

Question: The review mentions the "self-adjuvant effect" of circular RNA or mRNA lipid nanoparticles, but does not elaborate enough on the potential risks and safety concerns with these adjuvants. Adjuvants can play a key role in enhancing the immunogenicity of the vaccine, but they can also cause unwanted side effects or adverse reactions in some people. A more in-depth discussion of the safety of adjuvants would provide a more balanced perspective.

Answer: the adjuvant effect has been deeply discussed from “line 88 to 102”.

Question: Although mRNA vaccines have shown promising results in the fight against the COVID-19 pandemic, the reality is that they have not been able to definitively control and eliminate community transmission. This aspect should be noted in a section of Introduction to the article.

Answer: we all agree that in the reality, SARS-CoV-2 mRNA vaccines have not control and eliminate community transmission and we have added several reasons on this point in the revised paper from “line 124 to 139. “

Specific comments

Question: The authors should further explain that developing and testing vaccines against new variants can be a complex and time-consuming process, and better explain how RNA vaccine technology can effectively respond to these challenges.

Answer: This part has been explained from “line 287 to 307 and line 311 to 315.”

Question: The review not explore the potential cost barriers that could limit access to these vaccines, particularly in resource-limited settings. RNA vaccines can be more expensive to produce and distribute, which might affect their availability in low-income countries and raise equity concerns.

Answer: This part has been discussed from “line 635 to 638.”

Reviewer 3 Report

Major Comments

1.       This review article is not sufficiently referenced.  For example, lines 116-118 regarding “functional traits” of circRNA.  This paragraph should have several references but does not reference any publications.  Lines 148-175 comprise two paragraphs and contain zero references.  The authors must take the time to insert references as appropriate throughout the manuscript. 

Minor Comments:

1.       Line 13 – While SARS-CoV-2 emerged in 2019, the outbreak was not pandemic until 2020.

2.       Line 17 – What are “latent microbes”?  “Latent” is a very specific word and often applied to a very specific biological function of some large DNA viruses.  I think its use here may not be the best.  Please check. 

3.       Line 36 – “alternating” is the wrong word for this usage.  I think you mean “in 1971 and 1974, respectively.”

4.       Line 47 – “alternatively” is not the right word here either.  How could Moderna have been established in 2008 and 2010?

5.       Line 48 and throughout – Maybe spell out “message” when it is the first word in a sentence?  I’m not sure of the grammar rule for this point.

6.       Line 51 – “Rnase” is typically shown as “RNase”

7.       Line 79 – Spell out the genus and species “brucellosis.”  Italicize genus and species for both bacteria.

8.       Lines 84-85 – Please include a reference to support this statement about genome integration and host immune burden.

9.       Line 101-103 – We now have an approved vaccine for RSV.  Also, was there ever evidence that ADE was a road block to the development of a safe and effective ZIKV vaccine?

10.   Line 250 – SARS-CoV-2 emerged in December, 2019.  It did not result in 768 million cases and 69 million deaths by that month.  Please correct.

11.   Figure 2 is not mentioned in the text.  Viral evolution is wandering a bit far from the article topic.

Author Response

Major Comments

Quesition: This review article is not sufficiently referenced.  For example, lines 116-118 regarding “functional traits” of circRNA.  This paragraph should have several references but does not reference any publications.  Lines 148-175 comprise two paragraphs and contain zero references.  The authors must take the time to insert references as appropriate throughout the manuscript. 

Answer: We have gone through the paper and another 23 publications were inserted including all the suggestions you had in circRNA part. Thanks a lot for your professional input and comments. We have corrected all.

Minor Comments:

Question 1: Line 13 – While SARS-CoV-2 emerged in 2019, the outbreak was not pandemic until 2020.

Answer: We have corrected this. Please check it in the revised paper. 

Question 2: Line 17 – What are “latent microbes”?  “Latent” is a very specific word and often applied to a very specific biological function of some large DNA viruses.  I think its use here may not be the best.  Please check. 

Answer: We have substituted the word “latent” with “immunotolerant”

Question 3: Line 36 – “alternating” is the wrong word for this usage.  I think you mean “in 1971 and 1974, respectively.”

Answer: We have rewritten this sentence.

Question 4: Line 47 – “alternatively” is not the right word here either.  How could Moderna have been established in 2008 and 2010?

Answer: We have rewritten this sentence.

Question 5: Line 48 and throughout – Maybe spell out “message” when it is the first word in a sentence?  I’m not sure of the grammar rule for this point.

Answer: We have added the abbreviation to the “m” as messenger when the word mRNA first appears in “line 10.”

Question 6: Line 51 – “Rnase” is typically shown as “RNase”

Answer: “Rnase” has been corrected to “RNase.”

Question 7: Line 79 – Spell out the genus and species “brucellosis.”  Italicize genus and species for both bacteria.

Answer: This has been corrected in “line 80.”

Question 8: Line 101-103 – We now have an approved vaccine for RSV.  Also, was there ever evidence that ADE was a road block to the development of a safe and effective ZIKV vaccine?

Answer: For “was there ever evidence that ADE was a road block to the development of a safe and effective ZIKV vaccine?”. ADE is very common in Flaviviridae since ADE was discovered when scientists were developing Dengue vaccines. ZIKV, WNV, YFV, JEV, LGTV and TBEV generated antibodies sometimes may have cross reaction to other flavivirus since they have high similarity among the E proteins, usually presented as prM-E, a precursor form. Several publications have proved that JEV or dengue immunization may cause ADE of ZIKV when pre-clinical research was conducted. On the other hand, when developing ZIKV vaccine, Dengue and other flavivirus must be tested in ADE assay due to the concerns from the public.

Question 9: Line 250 – SARS-CoV-2 emerged in December, 2019.  It did not result in 768 million cases and 69 million deaths by that month.  Please correct.

Answer: This has been corrected “in line 282.”

Question 10: Figure 2 is not mentioned in the text.  Viral evolution is wandering a bit far from the article topic.

Answer: Figure 2 has been mentioned in “line 286.” The reason we discussed on the viral evolution is to emphasize the mutation rate of SARS-CoV-2, trying to give a more comprehensive discussion in the part “Coronavirus”.

Round 2

Reviewer 3 Report

The authors have adequately addressed my concerns.